# Assessment of Micro- and Nanoplastic Composition (Polymers and Additives) in the Gastrointestinal Tracts of Ebro River Fishes

**DOI:** 10.3390/molecules28010239

**Published:** 2022-12-28

**Authors:** Maria Garcia-Torné, Esteban Abad, David Almeida, Marta Llorca, Marinella Farré

**Affiliations:** 1Institute of Environmental Assessment and Water Research (IDAEA-CSIC), C/Jordi Girona, 18-26, 08034 Barcelona, Spain; 2GRECO, Institute of Aquatic Ecology, University of Girona, Campus Montilivi, 17003 Girona, Spain; 3Department of Basic Medical Sciences, School of Medicine, Universidad San Pablo-CEU, CEU Universities, Urbanización Montepríncipe, 28668 Boadilla del Monte, Spain

**Keywords:** microplastics, nanoplastics, plastic additives, fish, LC-HRMS, suspect screening, quantification

## Abstract

One of the main routes of fish exposure to micro- and nanoplastics (MNPLs) is their ingestion. MNPLs can act as reservoirs of organic contaminants that are adsorbed onto their surfaces, or that can leach from their complex formulations, with potential impacts on biota and along the aquatic food chain. While MNPLs have been reported in fishes worldwide, complete information on MNPL compositions, polymers and additives continues to be scarce. In this work, the presence of MNPLs in the gastrointestinal tracts (GIT) of fish from the Ebro River (Spain) was investigated using a double suspected screening approach to assess and quantify polymers and additives. The sample-preparation procedure consisted of sequential alkaline and acidic digestions with KOH and HNO_3_, followed by ultrasonic-assisted extraction (USAE) with toluene. The analysis of polymers was carried out with size-exclusion chromatography followed by high-resolution mass spectrometry using an atmospheric pressure photoionization source, operating in negative and positive ionisation modes (SEC-(±)-APPI-HRMS) using full-scan acquisition (FS). Plastic additives were assessed using high-performance liquid chromatography with a C18 analytical column coupled to HRMS equipped with an electrospray ionisation source operating under positive and negative conditions (LC-(±ESI)-HRMS). The acquisition was performed in parallel with full-scan (FS) and data-dependent scan (ddMS^2^) modes, working under positive and negative ionisation modes. The polymers most frequently detected and quantified in fish GITs were polysiloxanes, polyethylene (PE), polypropylene (PP) and polystyrene (PS). PE was detected in 84% of the samples, with a concentration range from 0.55 to 3545 µg/g. On the other hand, plasticisers such as phthalates and stabilisers such as benzotriazoles were the most frequently identified plastic additives.

## 1. Introduction

In recent years, plastic production and usage have continuously increased, and as they grow, their impact on aquatic ecosystems also increases [1]. The entrance of plastic litter into aquatic systems is due to different causes, such as ineffective waste management (deposition in landfills), wind events, littering on shorelines or at sea, loss of fishing tackle, stormwater runoff and treated wastewater discharges still containing microplastics (MPLs), among others factors [2,3].

MPLs are known as plastic particles smaller than 5 mm, while nanoplastics (NPLs) are those below 1 µm [4]. MPLs can be classified according to their origin, with primary ones entering the environment with particles below 5 mm, and secondary MPLs resulting from fragmentation and erosion in the environment of bigger plastic items [1,5].

Some of the main problems that derive from micro- and nanoplastic (MNPL) pollution are their contribution to eutrophication processes [6,7]; entrance into the aquatic food chain [8,9]; physical toxicity [10] contributing to oxidative stress and inflammatory lesions [11]; chemical toxicity (because of leaching monomers of the plastic polymers and additives used in plastic formulation, such as plasticizers, flame retardants, fillers, UV-stabilizers, coating finishers, colourants, and metals, among others) [10]; and potential contribution to transferring other co-contaminants from surrounding waters accumulated onto their surface to biota, also known as the Trojan Horse effect [12,13,14]. It should be highlighted that MNPLs’ toxicity is directly related to their size and shape. The smaller the particles, the bigger the active surface areas that facilitate both leaching of monomers and additives and the adsorption and transfer of other contaminants present in the surrounding environments to biota. Moreover, plastic particles cannot be digested, but NPLs can be translocated through tissues and bioaccumulated. For these reasons, NPLs are the most relevant from the toxicological point of view. Nonetheless, there is a substantial gap in information regarding MNPLs’ accumulation in biota and, particularly, regarding NPLs because of technical limitations and difficulties in their analysis.

Analytical techniques used for the characterisation of the size and shape of plastic particles are, in general, based on scanning electron microscopy (SEM) [15], SEM coupled to energy-dispersive X-ray spectroscopy (SEM-EDS) [15,16,17] and SEM with energy-dispersive X-ray spectroscopy (ESEM-EDS) [15,17]. Nowadays, the most common techniques for the investigation of the polymeric composition of MPLs are those based on spectroscopies such as Fourier-Transform Infrared (FTIR) and Raman spectroscopies, which are combined with microscopies to assess the number of particles. However, these approaches are limited by the particle sizes [15,16]. Using spectroscopic techniques, the smallest particle sizes that can be investigated are about 1 μm for μ-FTIR and for μ-Raman, a bit less, with μ-FTIR being the most commonly used recently to investigate MPLs in fish GITs [18,19]. Other promising approaches are pyrolysis gas chromatography (Pyr-GC-MS) [20] and size-exclusion chromatography coupled to high-resolution mass spectrometry (SEC-HRMS) [21], which are not limited by the particle size [2,15] and can provide qualitative and, coupled to HRMS, quantitative information in terms of mass units.

Under this framework, the central goal of this study was to assess the occurrence of MNPLs in the gastrointestinal tracts (GITs) of fish from the Ebro River (NE of Spain). The MNPL extraction from the GITs was achieved with sequential alkaline and acidic digestion. Then, the identification of the bigger particle fraction was carried out using FTIR. For the smallest fractions, the polymeric composition was determined with ultrasonic-assisted extraction (USAE) with toluene followed by size-exclusion chromatography (SEC) coupled with high-resolution mass spectrometry using an APPI source in negative and positive conditions (APC-(±)APPI-HRMS) according to the approach previously developed in our group [1]. In parallel, the main groups of plastic additives in the samples were studied using liquid chromatography coupled to HRMS using an electrospray ionisation source (ESI) under positive and negative conditions (LC-(±ESI)-HRMS) in full-scan acquisition mode. A suspected screening approach was employed to tentatively identify the plastic additives of MNPLs in fish GITs. Among the tentatively identified compounds, a prioritisation was carried out and based on these results, a selection of the most relevant additives was made. Then, these additives were confirmed and quantified in the samples. To the authors’ knowledge, this is the first study reporting plastic contamination in GITs of fish from the Ebro River.

## 2. Results and Discussion

### 2.1. FTIR Results

Visual analysis was used to count particles and fibres. In total, 39.6% of the MPLs were fibres while 60.4% were fragments. The samples whose filters presented particles/fibres bigger than 20 μm were analysed with FTIR. The results showed that the composition of most of the fibres bigger than 20 μm was cellulose (64%), 15% of the analysed fibres were polyester, 12% were of PE, 5% were PET, and 4% PP. In Appendix A, an example of a cellulose fibre is shown. Some fibres were not identified by FTIR due to their thickness. The observations show that there were, on average, 29 fibres/sample and 360 black particles/sample. Carnivorous and omnivorous fishes showed a higher presence of particles than detritivorous fishes, corresponding to the fact that fish that have a predatory diet (i.e., elevated trophic level) may accumulate and present higher levels of contamination than those that eat detritus [22] The results can be seen in Appendix A.

### 2.2. Analysis of MNPL Polymers

A total of 40 GIT samples of fishes collected in the last part of the Ebro River were analysed. In Figure 1, a summary of the polymers and additives are shown. As can be seen in Figure 1, only one sample was free of MNPLs. Therefore, no polymer contents or organic plastic additives were detected in it. This sample was CAT 2 (*Carassius auratus*), a small fish in the Cyprinidae family. Therefore, about 98% of studied fish were contaminated with MNPLs. Seven polymers were detected in the samples including PE, PI, PS, PP, PA, methacrylate and the inorganic polymer group, polysiloxanes. PE was the most frequently detected polymer in ca. 84% of the samples, followed by polyamide (PA) in 73% of the samples. Polysiloxanes were also in 75.6% of the samples, but in this case were identified at level 2 (not quantified). In Figure 2, the frequency of polymer detection is presented. As can be seen in Figure 3, the polymer at highest concentration was also PE, with concentrations ranging from 0.55 to 3546 µg/g. These results are in agreement with other studies in the same area analysing environmental matrices as water [1]. In general, the polymers found at higher concentrations in the environment are PP and PE. PE is mainly used in single-use plastic items and in packaging, while PP is used in building materials, packaging, and single-use items. On the other hand, polysiloxanes, also known as silicones, are the most important class of inorganic polymers. Silicones possess a fully inorganic backbone of -(Si-O)- repeating units. This type of polymer finds extensive use in both industrial and urban applications.

More evident differences can be seen by comparing the trophic level of the studied fishes (see Figure 3). Omnivorous fishes were those with a higher content in the GIT. In addition, the larger fish samples showed higher MNPL levels, as expected. In Appendix A some examples of chromatograms, mass spectra and KMD plots of identified polymers in the samples are provided.

### 2.3. Analysis of Plastic Additives

Raw data were processed with Compound Discoverer 3.1 software. The first step generated more than 50,000 potential compounds. Among them, 8095 structures were identified at level 5. Further filtration provided 1661 compounds identified at confidence level 4; among them, 1340 compounds were tentatively identified at level 3. The comparison with MS/MS spectrum of the further fragmented compounds provided 563 compounds tentatively identified at level 2. The selection of plastic additives provided 157 polar plastic additives. At the same time, the rest of the structures were natural compounds or compounds that cannot be directly related to the plastic industry, such as pharmaceutical compounds (pain killers or antibiotics), personal care products (PCPs), antibacterial or antifungal) and other contaminants in the aquatic environment, which probably were adsorbed onto the plastic surface. The different groups of compounds can be seen in Figure 4.

Plastic additives were identified in 95% of the samples. Among the different types of plastic additives, only medium- to non-polar compounds were detected because of the analytical method used. Plasticizers were the most-found compounds (present in 80% of the samples and 82% of the samples showing the presence of MNPL particles). Adipates and phthalates were the most common groups, and among them should be highlighted the presence of adipic acid, dimethyl phthalate, mono(2-ethylhexyl) phthalate (MEHP) and phthalimide, respectively. Another important group was the stabilizers, among which should be highlighted N, N-Bis(2-hydroxyethyl) dodecanamide, which is used as an anti-caking agent, and was present in 25% of the samples. Another frequently tentatively identified compound was 1,2-Benzisothiazolin-3-one, used as a preservative, which was in 17% of the samples. The groups of lubricants, colorants, and flame retardants were detected at lower rates (20%, 12%, and 7%, respectively), probably because these compounds are not properly extracted and analyzed under these conditions. In this case, another isolation and analytical approach would be required. Furthermore, some of them could require a more targeted analysis to meet the sensitivity requirements and conditions for their extraction and analysis.

In addition to plastic additives, other compounds detected in the samples were natural products and other anthropogenic-origin compounds that cannot be associated with plastic formulations. These contaminants were probably present in surrounding waters and some proportions were probably adsorbed onto plastic particles, before their ingestion by fish. Among these contaminants, some pharmaceutical compounds, such as primobolan (24%), which is an anabolic steroid; sarsasapogenin (17%), used in the synthesis of steroids; or citalopram hydrobromide (15%), used as an antidepressant, were detected in the samples. Another relevant group of detected compounds was PCPs such as civetone (63%), used as a perfume fixative and as a flavouring; undecanoic acid (17%), used as an antifungal agent; or compounds used as biocides such as N, N-Diethyl-meta-toluamide (DEET) (63.4%), used as an active compound in insect repellents.

### 2.4. Prioritisation of Plastic Additives

In Appendix A, the scoring parameters data of the tentatively identified compounds is presented. The 38 compounds showing the highest ranking are presented in Table 1.

Among them those presenting the highest frequency of detection and according to the standards’ availability, a selection of 20 compounds was confirmed with external standard curves and quantified in the samples. In Table 2, the compounds confirmed and quantified are shown.

Oleic acid was found to be the compound with the highest concentration of 28,000 ng/g. It is noticeable that oleic acid is used as an intermediate in polymer synthesis as well as a plasticizer; however, it is also one of the most common fatty acids in nature. Similarly, nonanoic acid reached 3000 ng/g. It is used as an intermediate in polymer synthesis, but it also occurs naturally. 4-methylphenol also occurs naturally; however, it has not been noticed in fish or aquatic environments, so it can be considered that it came from an anthropogenic source. Tributyl phosphate, laurolactam, tris(2-chloroethyl) phosphate, dimethyl phthalate and diacetone acrylamide come from anthropogenic sources and their concentrations ranged between 3 and 430 ng/g.

## 3. Materials and Methods

### 3.1. Study Area and Field Sampling

The fish samples were collected at two locations from Catalonia, Xerta (40°54′14″ N, 0°29′36″ E) and Tortosa (40°48′56″ N, 0°31′14″ E), both in the last part of the Ebro River. The climate in the lowland of the River Ebro is typically Mediterranean, with rainfall concentrated in autumn and spring (200–300 mm) and intense summer drought (<50 mm). The flow regime is pluvio-nival because of the left-margin tributaries from the Pyrenees. The average annual temperature ranges between 10 and 15 °C. The lowest temperatures occur in winter and can reach down to 0 °C, while the highest in summer can reach over 40 °C. The substratum in the area is mainly calcareous, with Cenozoic limestones, gypsum and alluvial sediments. The sampling campaign was carried out in April 2019, the peak season for rice and crop activities and the initiation of touristic activities in this area (i.e., the annual peak for the highest level of water pollution).

A total of 40 fish samples was collected, 20 from each sampling site. Specifically, fish species collected were *Alburnus alburnus* (6), *Cyprinus carpio* (2), *Carassius auratus* (6), *Dicentrarchus labrax* (3), *Ictalurus punctatus* (2), *Liza* sp. (6), *Luciobarbus graellsii* (1), *Mugil cephalus* (5), *Pseudorasbora parva* (1), *Rutilus rutilus* (2), *Silurus glanis* (1) and *Squalius laietanus* (5). Appendix A provides a summary of the samples. We selected native species that are abundant in each river transect. This facilitated the capture and guaranteed the representativeness of the sample. Moreover, all selected fishes were large enough to facilitate the GIT extraction and sample manipulation. Furthermore, they were species that did not present any cataloguing in conservation matters, so the sampling did not inflict any damage on the viability of wild populations at risk. Fish were collected from all meso-habitats present in the river (e.g., runs, riffles and pools), from both the left and right margins and at each sampling site (500 m river length), a representative sub-sample was collected to encompass the environmental variability. Sampling was performed by electrofishing from a boat (4.5-m aluminium hull) using a 2000 W DC generator at 1000 V and 16 A (Model: 5.0-GPP Smith-Root Inc., Vancouver, WA, USA), along with dip nets (2.5 m long pole, 50 cm diameter net, 10 mm mesh size). Two anodes were suspended from booms and mounted on the boat’s bow, and a cathode was mounted along each side of the hull. A single pass was made following a zigzagging and upstream direction without using block nets in every sampling site. After each survey, fish were identified to the species level and counted. Then, a fish sub-sample was immersed in an overdose solution of anaesthetic (MS-222) for 15 min. Euthanized fish were measured for fork/total length (FL/TL, ±1 mm), weighed (wet body weight, ±0.1 g), labelled, stored and frozen (−20 °C) on the same date of collection (less than 2 h since euthanasia) until laboratory processing. The remaining individuals of non-native species were euthanized according to the same procedure described above. On the other hand, native species were kept in a tank with supplied oxygen (two battery-operated aerators with portable pumps) until fully recovered before being released. All field procedures complied with Europe and Spain’s animal use and care regulations (specific licences were granted for Scientific Field Research in the River Ebro). Fish were collected by trained personnel. Thus, no adverse effects were caused to the wildlife in the study habitats and all native fish fully recovered from the anaesthetic.

### 3.2. Chemicals and Reagents

Polymer analytical standards for GPC/SEC were acquired for polypropylene (PP, Mw 1175 Da; Mn 1120 Da; PDI [Mw/Mn] 1.04) from American Polymer Standards Corporation (Mentor, OH, USA). For polyisoprene (PI, Mw 1000 Da; Mn 854 Da; PDI [Mw/Mn] 1.17) and polybutadiene (PBD, Mw 1000 Da; Mn 925 Da; PDI [Mw/Mn] 1.08) STD KIT (from Waters Cromatografía (Cerdanyola del Vallès, Barcelona, Spain), and polyethylene (PE) and polystyrene (PS) (Mw 1220 Da; Mn 1120 Da; PDI [Mw/Mn] 1.09) provided by Polymer Standard Service GmbH (PSS, Mainz, Germany).

Nitric acid (65% purity), NaCl (99% purity), and KOH (99.99% purity) were provided by Sigma-Aldrich (Steinheim, Germany).

Methanol and water (HPLC grade) were both purchased from LiChrosolv (Darmstadt, Germany). Toluene (HPLC grade) was supplied by SupraSolv (Darmstadt, Germany).

Glass microfiber filters of 0.70 µm pore size were acquired from Whatman (GF/F, Whatman TM, Maidstone, UK).

### 3.3. Sample Pre-Treatment

The first step in sample preparation consisted of milling the fish’s GITs. Next, the frozen samples (−20 °C) were chopped with an agate mortar with liquid nitrogen and finely ground in a cryogenic grinder (6875^®^ Freezer/Mill) in 3 cycles of 1 min each. Then, a digestion process was used to eliminate the organic matter according to the protocol developed by Schirinzi et al. [9]. Very briefly, the samples were weighted in glass vials, filled with a brine solution and mixed with a vortex. The mixtures were then transferred to a glass beaker with 100 mL of NaCl (120 g/L), homogenized and left to rest for an hour. Then, the supernatant was filtered through 0.45 µm filters using a vacuum pump and the filters were submerged in 50 mL KOH 10% overnight. After this time, the filters were recovered and used to filter the KOH solution using a vacuum pump. The filters were treated with 40 mL HNO_3_ 20% to remove biological material. The final filters were kept in glass Petri dishes and dried in the extraction hood overnight. The dry filters were observed with an optical microscope (×100) to identify fibres and bigger particles, and the different shapes and colours were registered. Finally, each filter was weighted and divided into one half and two quarters.

The first half was used to analyse polymers of MNPLs using APC-(±)APPI-HRMS, another quarter was used for the analysis of plastic additives in the MNPLs with LC-(±)ESI-HRMS, and the other quarter for the analysis of MPLs bigger than 5 µm with FTIR.

***Extraction of polymers for APC-(±)APPI-HRMS analysis.*** Half of the filter sample was introduced in a glass beaker and extracted with 10 mL toluene for 10 min using USAE. Then, the supernatant was collected and transferred to an amber 50 mL vial. This process was repeated twice, and the final supernatant (30 mL) evaporated to approximately 1 mL under a gentle nitrogen stream and a temperature of 40 °C. Then, the extract was transferred to an LC vial, evaporated under a nitrogen stream and reconstituted to 0.5 mL with toluene. The final extracts were kept at −20 °C until analysis.

***Extraction of plastic additives for LC-(±)ESI-HRMS analysis.*** A quarter of each filter was extracted using USAE with 10 mL of methanol per 10 min. The same process was repeated twice, and the extracts were combined. Then, the solvent was evaporated to approximately 1 mL under a gentle nitrogen stream and transferred to an LC vial. Finally, the samples were centrifuged (10 min at 4000 rpm, 20 °C), and the supernatants were transferred into new LC vials, evaporated nearly to dryness, reconstituted in 0.5 mL (methanol: water, 1:9) and kept until analysis at −20 °C.

### 3.4. FTIR Analysis of MPL Polymers

The fibres and fragments visually identified with an optical microscope were analysed using ATR-FTIR (IR PerkinElmer Frontier, Akron, OH, USA). The instrumental parameters ranged from 4000 to 230 cm^−1^, 16 scans accumulated and 4 cm^−1^ of resolution.

### 3.5. LC-HRMS Analysis of MNPL Polymers

Polymer analysis was based on Llorca et al. [1], with minor modifications. The chromatographic separation was performed using an Acquity chromatograph (Waters, Milford, MA, USA) system equipped with an advanced polymer chromatography column SEC (ACQUITY APC^TM^ XT 45 1.7 μm 150 mm) and toluene as mobile phase in isocratic conditions. The chromatograph was coupled to a Q-Exactive hybrid quadrupole-Orbitrap mass spectrometer (Thermo Fisher Scientific, San Jose, CA, USA) equipped with an APPI ionisation source operating in positive and negative conditions, by separate injections. The flow rate was kept at 0.5 mL/min during the 5 min of the analysis. The injection volume was 10 µL. The acquisition was performed in full-scan mode (*m*/*z* 500–3000) working at 17,500 full width at half maximum (FWHM) resolution. For the APPI, sheath gas was set at 60 a.u., auxiliary gas at 35 a.u. and S-lens RF at 100 a.u. The column temperature was kept at 40 °C to facilitate the separation while the injector was kept at 16 °C to avoid solvent evaporation.

The data were manually processed with Xcalibur 2.1 Qual Browser software.

### 3.6. LC-HRMS Analysis of Plastic Additives

The chromatographic separation to analyse the additives in the extracts of MNPLs was achieved with a C18 analytical column (Purospher^®^ STAR RP-18 end-capped column (3 μm, 2.1 × 50 mm) from Merck) coupled to HRMS equipped with an ESI source, working in positive and negative ionization conditions by separate injections. The mobile phase consisted of (A) HPLC-water and (B) acetonitrile (in negative mode) or HPLC-water acidified with 0.05% formic acid (in positive conditions). The elution gradient conditions for the LC mobile phase started with 90% eluent A holding for 2 min and decreasing to 10% in 8 min, holding for two more min and rising to initial conditions (90% A) in one min and, finally, the re-equilibration of the system was achieved in 2 min. The flow rate was kept at 0.2 mL/min throughout the total chromatographic run of 15 min. The sample injection volume was set at 10 μL. Data were acquired in full scan (90–1500 Da) with an FWHM of 70,000 and, in parallel, in data-dependent scan at a resolution of 35,000 FWHM where the 10 most intense ions from the first full scan were further fragmented with an isolation of 1.0 Da and with a collision energy of 30 a.u.

The whole data were processed using Compound Discoverer 3.1.

### 3.7. Data Treatment for MNPL Polymers

The total ion chromatograms (TIC) obtained using full-scan (FS) acquisition using the Xcalibur (Thermo Fisher Scientific) software were processed.

Peak picking and grouping were used as the first screening step. The TIC was interrogated every 30 s by intervals of 500 Da where peaks which mass spectra showed repetitive mass losses were marked as suspected polymers. The next filtration step was considering the intensity and, when the peak area was less than three times larger than the maximum peak area in the blanks, data were discarded as background. Then, selected profiles were tentatively identified using the repetitive mass loses with an error mass within 10 ppm. The first list of suspected polymers was created (level 4) and it was subsequently filtered by comparison with a homemade polymers library that included the 100 most-used ones [1] that can be slightly soluble in toluene (level 3). The Kendrick Mass Defect (KMD) was used for further filtering and to tentatively identify the polymers (identification level 2). KMDs were calculated according to equations described elsewhere [21], taking in each case PolymerX/(round Polymer X-2) as base unit. Some of the identified polymers (PP, PE, PS, PI) have also been quantified; comparison with their standards allowed us to achieve identification at level 1.

### 3.8. Data Treatment for Plastic Additives and Potential Co-Contaminants Adsorbed to Plastic Particles’ Surfaces

Raw data were processed with Compound Discoverer 3.1 software. The first step included the data alignment, the comparison with control samples (filter blank analysed with each set of samples) and the comparison with available online and in-house databases. The first automatic processing generated thousands of potential compounds. Then, all the data were filtered using a series of restrictions such as the ratio sample/blank higher than 1.5, a minimum area of 1,000,000 a.u., retention time higher than 1 min and coincidence with expected mass spectra higher than 80%. This list of compounds was filtered using Visual Basic for Applications. The suspects with an error mass within ±2.5 ppm and present in the three replicates were identified at confidence level 5. Then, suspects from positive and negative ionisations were put together and the potential molecular formulae compared with data available in the ChemSpider and mz Cloud databases to achieve a tentative identification at confidence level 4. Further filtering steps consisted of comparing isotopic patterns, providing identification at confidence level 3. To obtain the tentative identification (level 2), the productions obtained from the MS/MS spectrum of a suspected compound were compared with the spectrum of a standard or a predicted fragmentation pattern using the information contained in the online databases.

### 3.9. Prioritization of Plastic Additives and Confirmation and Quantification of Selected Compounds

The prioritization of additives according to their potential impact was performed following the previous procedure [23] with minor modifications. The scored parameters comprehend the detection frequency in percentage, biodegradability (half-life), Log BAF (bioaccumulation factor) and toxicity values based on the 50% lethal dose (LD_50_) laboratory test in rats by oral administration. The score ranged between 0 and 100 for each parameter, and 100 points were added if carcinogenicity, mutagenicity, or reproductive toxicity were already reported. Thus, 500 was the maximum score. In Table 3, the parameterisation and scoring are shown. It is noteworthy that parameters such as biodegradation (half-life) and bioaccumulation factor (BAF) were predicted using EPI Suite^TM^ software (US EPA. [2022]. Estimation Programs Interface Suite^TM^ for Microsoft^®^ Windows, v 4.11. United States Environmental Protection Agency, Washington, DC, USA.)

Among the compounds with maximum scores detected at tentative level 2, those with higher frequencies and standards available were confirmed with external standard curves and were quantified in the samples.

### 3.10. Quality Assurance and Quality Control

To avoid polymer contamination, cotton clothes and laboratory coats were used, as well as nitrile gloves. Before usage, all material was rinsed with water and ethanol. Samples were covered with aluminium foil while digestion processes occurred. Blanks consisting of pure solvent were also analysed to define the background noise.

## 4. Conclusions

The results of this study show that 98% of the samples were contaminated with MNPLs. The results of polymer composition with FTIR for the range of big particle size and with SEC-(APPI)-HRMS for the smallest ones were consistent. SEC-(APPI)-HRMS showed that the most frequently detected polymers in GITs of fish were PE, PP, PS and polysiloxanes, in agreement with other works. PE was detected in 84% of the samples and was also found at higher concentrations ranging from 0.55 to 3545.54 µg/g.

The presence of plasticisers such as phthalates and adipates should be highlighted among plastic additives. Among stabilisers, N, N-Bis(2-hydroxyethyl) dodecanamide and 2,5-di-tert-Butylhydroquinonewere some of the most frequently found.

The results of this study highlight the complexity of plastic contamination and the necessity for integrative approaches to assess more realistically the potential impact of MNPL pollution in aquatic ecosystems and through the food web on human health.

This study showed that suspected screening approaches are effective procedures to obtain a comprehensive vision of plastic contamination in complex matrices and inform about the potential transfer of this type of contamination to the food chain. Moreover, it provides information about the most recalcitrant groups of plastic contaminants and, therefore, those that should be closely studied.

## Figures and Tables

**Figure 1 molecules-28-00239-f001:**
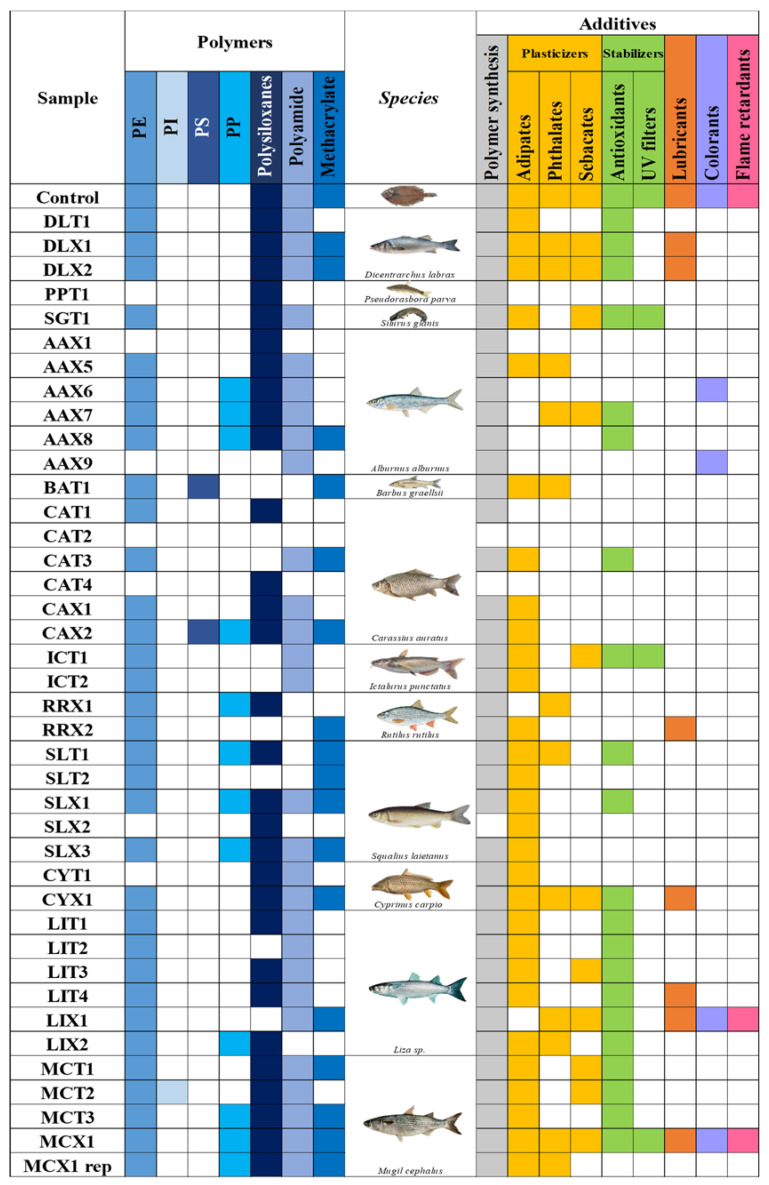
Summary of the polymers and plastic additive groups found in the GIT samples.

**Figure 2 molecules-28-00239-f002:**
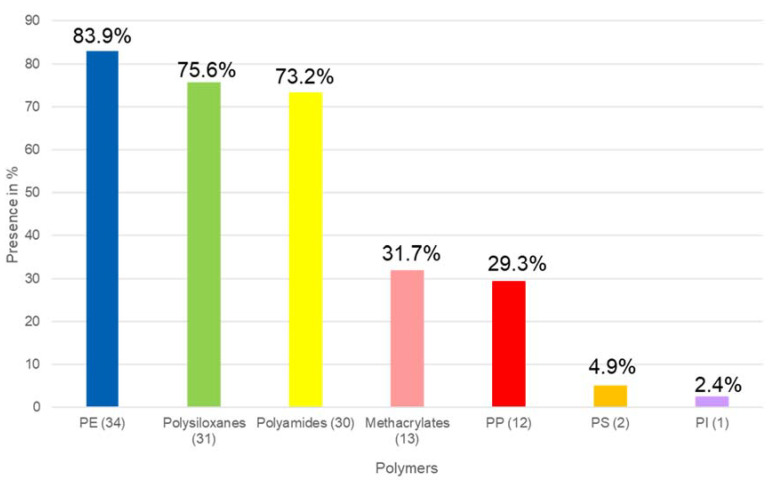
Frequency of polymer detection in the GIT samples.

**Figure 3 molecules-28-00239-f003:**
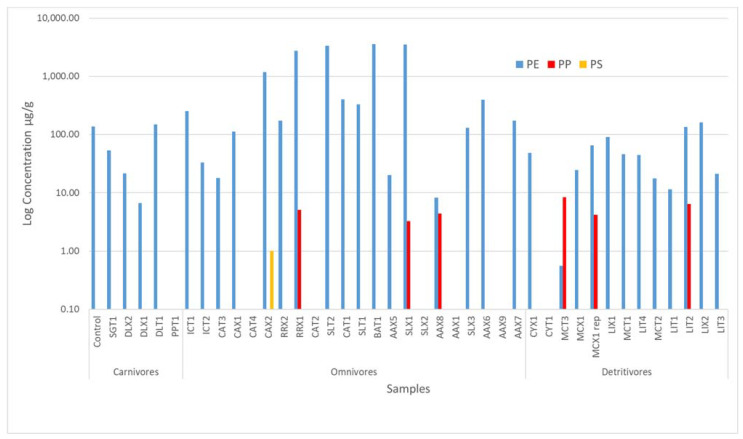
Concentrations of polymers in the samples expressed in equivalent concentrations of µg/g.

**Figure 4 molecules-28-00239-f004:**
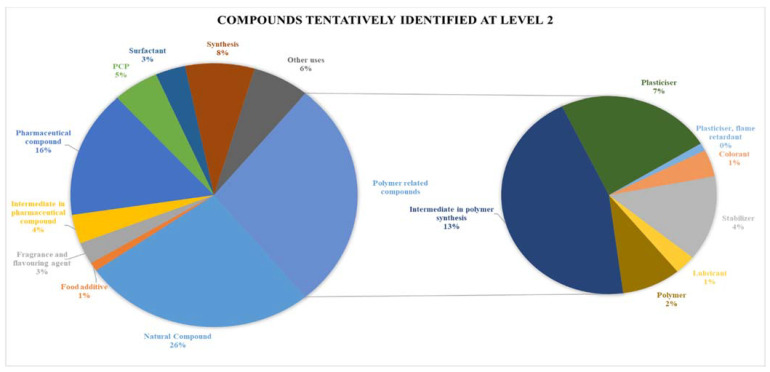
Compounds and plastic additives classification.

**Table 1 molecules-28-00239-t001:** Final ranking for the 5 highest-punctuation compounds.

Position	Compounds	Final Score
1	Oil Red O	225
1	5-methyl-2,2′-(1,2-Ethenediyldi-4,1-phenylene)-Bisbenzoxazole	225
1	Oleic acid	225
1	3-Vinyl-7-oxabicyclo[4.1.0]heptane	225
2	Isopropyl myristate	200
2	2,6-di-tert-butyl-4-ethylphenol	200
2	Dodecamethylpentasiloxane	200
3	2,5-di-tert-Butylhydroquinone	175
3	2,6-Dicyclohexylphenol	175
3	2,2′-Biquinoline	175
3	Acridine	175
3	Trihexyl 1,2,4-benzenetricarboxylate	175
3	Tetramethylurea	175
4	Citroflex A-4	150
4	Diisobutyl adipate	150
4	3-(3,5-di-tert-butyl-4-hydroxyphenyl) propanoic acid	150
4	dehydroabietic acid	150
4	2-tert-Butyl-4-methoxyphenol	150
4	Mono(2-ethylhexyl) phthalate (MEHP)	150
4	2,6-di-tert-Butyl-4-methoxyphenol	150
4	BIS T-BUTYLDIOXYISOPROPYLBENZENE	150
4	4-Toluic acid	150
5	1,2-Benzisothiazolin-3-one	125
5	bisphenol A diglycidyl ether	125
5	Tetrahydrofurfuryl acrylate	125
5	1,1-bis(tert-butylperoxy)-3,3,5-trimethylcyclohexane	125
5	4-Nonylphenol	125
5	3,3,5-Trimethylcyclohexyl methacrylate	125
5	Hexamethyldisiloxane	125
5	2-Ethoxyethyl acrylate	125
5	Dicyclohexylamine	125
5	o-tert-Octylphenol	125
5	Decyl octyl phthalate	125
5	vinyl propanoate	125
5	Bis(3-methoxypropyl) adipate	125
5	3-Vinyltoluene	125
5	Suberic acid	125
5	12-Crown-4	125

**Table 2 molecules-28-00239-t002:** Concentration values for the highest-scored compounds.

Sample	4-Nonylphenol	Oleic Acid	Decyl Octyl Phthalate *	Mono(2-Ethylhexyl) Phthalate (MEHP) *	Bisphenol A Diglycidyl Ether *	2,6-Dicyclohexylphenol *	2,6-di-tert-Butyl-4-Ethylphenol *	2,6-di-tert-Butyl-4-Methoxyphenol *	2-tert-Butyl-4-Methoxyphenol *	Isopropyl Myristate *	2-Ethoxyethyl Acrylate *	Tetrahydrofurfuryl Acrylate *	o-tert-Octylphenol *	3,3,5-Trimethylcyclohexyl Methacrylate *	Vinyl Propanoate *	Dodecamethylpentasiloxane *	Hexamethyldisiloxane *	3-(3,5-Di-Tert-Butyl-4-Hydroxyphenyl)Propanoic Acid *	Dehydroabietic Acid *	Suberic Acid *
**Control**	3124			140.7						<LOQ	284.2				9686.0	315.7		63.87		
**DLT1**								1773				67.73								
**DLX1**				31.53	997.8		14,462			<LOQ	108.5	203.3			<LOQ	354.7	5259			
**DLX2**		15,115		74.81						<LOQ	87.83				<LOQ	227.7	924.2	38.24		
**SGT1**							1708	2362			147.3									
**AAX5**			186.9																	
**AAX7**			859.8																	
**AAX8**																				76.51
**CAT3**											498.4									
**ICT1**								900.4												
**ICT2**										<LOQ										
**RRX2**																				29.10
**SLT1**																		109.7		
**CYT1**												866.8								
**CYX1**		3096		106.4		572.9			5453	<LOQ		59.12	129.6			181.6	1342			
**LIT1**								10,173				678.1								
**LIT2**								2657	174.1		1163									
**LIT3**									249.7											
**LIT4**									56.29	<LOQ										
**LIX1**				99.64			30,747			<LOQ	172.5	<LOQ			<LOQ		1463		423.1	83.45
**LIX2**		28,319		147.7						<LOQ										
**MCT1**							1191										2191			
**MCT2**											1344									
**MCT3**							3003													
**MCX1**		6654		953.0			65,499			<LOQ	281.3	143.4			<LOQ		1765		667.3	
**MCX1 rep**												41.09		32.48			1185			

* Compounds quantified using an equivalent standard.

**Table 3 molecules-28-00239-t003:** Scoring system for prioritisation of the tentatively identified substances.

Detection Frequency	Biodegradability *	Log BAF *	LD50 (mg/Kg)	Score
<5%	Days	<2	>1000	0
5–30%	Weeks	2–3	100–1000	25
30–50%	Weeks–Months	3–4	10–100	50
55–80%	Months	4–5	1–10	75
>80%	Recalcitrant	>5	<1	100

* Estimations using EPI Suite software (United States Environmental Protection Agency, US EPA).

## Data Availability

Not applicable.

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
