# Peer review of "Assessment of Micro- and Nanoplastic Composition (Polymers and Additives) in the Gastrointestinal Tracts of Ebro River Fishes"

_molecules, 2022, doi:10.3390/molecules28010239_

Round 1

Reviewer 1 Report

The figures are missing and the article should be fully revised before publication. See below for more details:

Line 95: what do you mean with “the most interesting”? This is not a rigorous sample selection criteria. Please justify. Also, what do you mean with “the most common spectrum”? Can you please provide quantitative data? How many fibers? How many were cellulosic? Etc.

Methods should be presented before the results and discussion section to ease the interpretation of this manuscript.

Fig 1, 2 and 3 are all missing. Where are the figures?

Lines 119-120: Please note that contrarily to what you say here, PP is also widely used in packaging, single-use and consumer items.

Also, regarding the results, the number of particles per invididuals should be reported for all species as well as the frequency of occurrence in each species separately. Please follow the best available reporting guidelines about microplastic ingestion studies. See for instance:
- Bessa, F., Frias, J., Kögel, T., Lusher, A., Andrade, J.M., Antunes, J., Sobral, P., Pagter, E., Nash, R., O’Connor, I. and Pedrotti, M.L., 2019. Harmonized protocol for monitoring microplastics in biota. Deliverable 4.3.
http://dx.doi.org/10.25607/OBP-821
- Cowger, Win, et al. "Reporting guidelines to increase the reproducibility and comparability of research on microplastics." Applied spectroscopy 74.9 (2020): 1066-1077.

Also, regarding the results, the number of particles extracted, the exact number of fibers, the exact number of cellulosic fibers etc, should all be reported unambiguously in the results section. How many fibers per individual? How many fibers were cellulosic fibers? Please note that cellulosic fibers cannot be classified as microplastics since they are not synthetic polymers and hence, they should be excluded from the results or presented separately from the synthetic particles. Please report the results more clearly.

Lines 199-200: please add total number of individuals sampled for each of these species.

Author Response

Thank you for your revision. Please find your comments in bold, and next are our responses. The changes in the new version of the manuscript have been highlighted in yellow.

Line 95: what do you mean with “the most interesting”? This is not a rigorous sample selection criteria. Please justify.

You are right, and it was changed in the new version of the manuscript. In fact, the paper is focused on small-range particles but in order to have some comparison with other studies using FTIR we have selected the samples with a higher number of particles bigger than 20 microns and were analyzed by FTIR

Also, what do you mean with “the most common spectrum”?

Most of the particles bigger than 20 microns were of cellulose, showing a common spectrum. This paragraph has been reformulated in the new version.

Can you please provide quantitative data? How many fibers?

This information is reported in table S1 and in the new paragraph of the FTIR results section. 

Methods should be presented before the results and discussion section to ease the interpretation of this manuscript.

The general order of the Molecules journal is Abstract, Introduction, Results, Discussion Material and methods, and Conclusions. But if this is finally free and you consider that it will be better, we will change it.

Fig 1, 2 and 3 are all missing. Where are the figures?

During submission, we provided a complete pdf; then the Molecules submission system created a new pdf in which something failed, and the figures were deleted. But in the new version, we inserted the figures in a different form to avoid this problem.

Lines 119-120: Please note that contrarily to what you say here, PP is also widely used in packaging, single-use and consumer items.

Reformulated as requested

Also, regarding the results, the number of particles per invididuals should be reported for all species (TableS1) as well as the frequency of occurrence in each species separately. Please follow the best available reporting guidelines about microplastic ingestion studies. See for instance:

- Bessa, F., Frias, J., Kögel, T., Lusher, A., Andrade, J.M., Antunes, J., Sobral, P., Pagter, E., Nash, R., O’Connor, I. and Pedrotti, M.L., 2019. Harmonized protocol for monitoring microplastics in biota. Deliverable 4.3. http://dx.doi.org/10.25607/OBP-821

- Cowger, Win, et al. "Reporting guidelines to increase the reproducibility and comparability of research on microplastics." Applied spectroscopy 74.9 (2020): 1066-1077.

Also, regarding the results, the number of particles extracted, the exact number of fibers, the exact number of cellulosic fibers etc, should all be reported unambiguously in the results section. How many fibers per individual? How many fibers were cellulosic fibers? Please note that cellulosic fibers cannot be classified as microplastics since they are not synthetic polymers and hence, they should be excluded from the results or presented separately from the synthetic particles. Please report the results more clearly.

We have used FTIR for a selected number of samples in which we found particles bigger than 20 microns. Still, the central aim of this work was to provide a quantification in terms of the mass of each polymer per sample of the particles between 700nm and 20 microns. To do it, we extracted them in toluene, and then we analysed them by LC-HRMS. For the smallest fraction, we cannot count and provide shapes…only the mass of each polymer. We have counted and analysed the particles bigger than 20 microns by FTIR, which is reported in table S1. 64% were cellulosic fibres; only 36 % were polymers. While for the smallest fraction, we only can report the polymers because cellulose cannot be extracted by toluene.

Lines 199-200: please add total number of individuals sampled for each of these species.

Added as requested.

Reviewer 2 Report

Is there any criteria followed in selection of fish species?

Represent scientific names in italics.

Conclusion is too lengthy.

Table 1 is cited after Tables 2 and 3.

Tables in supplementary materials are not cited in order in manuscript.

Table S4 is mentioned in manuscript. But, it is not cited in text and not available in supplementary information.

Tables 1 and 2 are not found in manuscript.

Figures 1 to 4 are not found in manuscript.

References from 2022 could not be found.

Author Response

Thank you for your comments. Please find the specific comments in bold below and our answers. On the other hand, the changes incorporated in the new version of the manuscript are highlighted in yellow.

Is there any criteria followed in selection of fish species?

The explanation was extended in the new version of the manuscript. Basically, as indicated in the first version of the manuscript, we have selected abundant native species of each river transect. This facilitates the capture and guarantees the representativeness of the sample. Moreover, all selected fishes were those with enough size to facilitate the GIT extraction and sample manipulation. And they were species that do not present any cataloguing in conservation matters, so the sampling does not imply any damage to the viability of wild populations at risk. Finally, we sampled different sites in the river to have representative species of all meso-habitats present in the river (lines 202-204).

Represent scientific names in italics.

Changed as requested

Conclusion is too lengthy.

The conclusions section has been shortened in the new version of the manuscript.

Table 1 is cited after Tables 2 and 3.

Corrected in the new version of the manuscript

Tables in supplementary materials are not cited in order in manuscript.

Corrected in the new version of the manuscript

Table S4 is mentioned in manuscript. But, it is not cited in text and not available in supplementary information.

Corrected in the new version of the manuscript

Tables 1 and 2 are not found in manuscript.

The manuscript was damaged during the submission, but we have changed the table and figures section to avoid new problems.

Figures 1 to 4 are not found in the manuscript.

The manuscript was damaged during the submission, but we have changed the table and figures section to avoid new problems.

References from 2022 could not be found.

Recent references have been included.

Reviewer 3 Report

The aim of the study was to assess the occurrence of micro and nanoplastics in the gastrointestinal tract of fish of the Ebro River (NE of Spain). The MS is interesting and relevant to the field. The experimental design is adequate and results are reliable. Please find below comments for minor revision: 

Abstract:

The phrase "The most frequently detected polymers found in GIT have been polysiloxanes, polyethylene (PE), polypropylene (PP), and polystyrene (PS) was detected in 84 % of the samples, with a concentration range from 0.55 to 3545 µg/g." is unclear. Please rewrite. 

Results: 

Line 94: "The samples whose filters were more interesting were further analysed by FTIR" Authors should clarify "more interesting". Do you have criteria? 

Line 125: “In addition, the higher the GIT, the higher the MNPLs concentration as expected”. What does it mean? Please clarify.

Conclusion:

Lines 362-371: Authors should consider relocating the first paragraph as the last. It could highlight core finds first and to finalize with recommendations.

Author Response

Thank you for your revision. Please find your comments in bold, and next are our responses. The changes in the new version of the manuscript have been highlighted in yellow.

Abstract:

The phrase "The most frequently detected polymers found in GIT have been polysiloxanes, polyethylene (PE), polypropylene (PP), and polystyrene (PS) was detected in 84 % of the samples, with a concentration range from 0.55 to 3545 µg/g." is unclear. Please rewrite. 

Changes as requested

Results: 

Line 94: "The samples whose filters were more interesting were further analysed by FTIR" Authors should clarify "more interesting". Do you have criteria? 

Changed as requested

Line 125: “In addition, the higher the GIT, the higher the MNPLs concentration as expected”. What does it mean? Please clarify.

Changed as requested

Conclusion:

Lines 362-371: Authors should consider relocating the first paragraph as the last. It could highlight core finds first and to finalize with recommendations.

Changed as requested

Round 2

Reviewer 1 Report

Most issues were addressed by the authors during revision. Still, before publication I suggest to mention also the additives in the title of this manuscript.
Also, at the beginning of the results section, please clearly specify how many of the particles counted were fibers and if all particles analyzed by FTIR were fibers? This is unclear and should be better described. 

Also, at line 96: replace "were analysis" with "were analysed"? 

Author Response

Thank for your fast reply. Below your suggestions please find our responses.

Most issues were addressed by the authors during revision. Still, before publication I suggest to mention also the additives in the title of this manuscript.

Additives have been included in the title.

Also, at the beginning of the results section, please clearly specify how many of the particles counted were fibers and if all particles analyzed by FTIR were fibers? This is unclear and should be better described. 

Yes, the percentage of fragments and fibers has been included in the new version of the manuscript and more details of each particular sample can be found in table S1

Also, at line 96: replace "were analysis" with "were analysed"? 

Changed as requested

The correction/additions are highlighted in yellow in the new version of the manuscript.

Reviewer 2 Report

Manuscript has been revised well and looks good when compared to version 1. But, still, some minor queries have not been rectified.

Author Response

We hope that in this new version all previous queries have been corrected.

Thank you for your time.